# An Integrated Approach to Unravelling Smallholder Yield Levels: The Case of Small Family Farms, Eastern Region, Ghana

**Ibrahim Wahab [1,2,\*], Magnus Jirström [2] and Ola Hall [2]**

[1] Department of Geography and Resource Development, University of Ghana, Legon, Accra GA-489-1680, Ghana

[2] Department of Human Geography, Lund University, 223 62 Lund, Sweden; magnus.jirstrom@keg.lu.se (M.J.); ola.hall@keg.lu.se (O.H.)

[*] Correspondence: Ibrahim.wahab@keg.lu.se; Tel.: +46-73-764-5474

**Abstract:** Yield levels and the factors determining crop yields is an important strand of research on rainfed family farms. This is particularly true for Sub-Saharan Africa (SSA), which reports some of the lowest crop yields. This also holds for Ghana, where actual yields of maize, the most important staple crop, are currently about only a third of achievable yields. Developing a comprehensive understanding of the factors underpinning these yield levels is key to improving them. Previous research endeavours on this frontier have been incumbered by the mono-disciplinary focus and/or limitations relating to spatial scales, which do not allow the actual interactions at the farm level to be explored. Using the sustainable livelihoods framework and, to a lesser extent, the induced innovation theory as inspiring theoretical frames, the present study employs an integrated approach of multiple data sources and methods to unravel the sources of current maize yield levels on smallholder farms in two farming villages in the Eastern region of Ghana. The study relies on farm and household survey data, remotely-sensed aerial photographs of maize fields and photo-elicitation interviews (PEIs) with farmers. These data cover the 2016 major farming season that spanned the period March–August. We found that the factors that contributed to current yield levels are not consistent across yield measures and farming villages. From principal component analysis (PCA) and multiple linear regression (MLR), the timing of maize planting is the most important determinant of yield levels, explaining 25% of the variance in crop cut yields in Akatawia, and together with household income level, explaining 32% of the variance. Other statistically significant yield determinants include level of inorganic fertiliser applied, soil penetrability and phosphorus content, weed control and labour availability. However, this model only explains a third of the yields, which implies that two-thirds are explained by other factors. Our integrated approach was crucial in further shedding light on the sources of the poor yields currently achieved. The aerial photographs enabled us to demonstrate the dominance of poor crop patches on the edges and borders of maize fields, while the PEIs further improved our understanding of not just the causes of these poor patches but also the factors underpinning delayed planting despite farmers' awareness of the ideal planting window. The present study shows that socioeconomic factors that are often not considered in crop yield analyses—land tenure and labour availability—often underpin poor crop yields in such smallholder rainfed family farms. Labour limitations, which show up strongly in both in the MLR and qualitative data analyses, for example, induces certain labour-saving technologies such as multiple uses of herbicides. Excessive herbicide use has been shown to have negative effects on maize yields.

**Keywords:** smallholder agriculture; yield determinants; socioeconomic factors; maize; principal component analysis; photo-elicitation interviews; Ghana

## 1. Introduction

Significant progress has been made in food production and productivity at the global level in the past half-century. In spite of this, producing adequate quantities to meet a growing demand is still largely a mirage in some developing regions, especially large parts of Sub-Saharan Africa (SSA). In this region, domestic food production has not been able to keep up with population growth [1]. To illustrate this stagnation in SSA agricultural production and productivity, estimates based on the Food and Agriculture Organization (FAO) data show that average cereal yields from the region was 57% of that of the world average in the 1960s, but reduced to 47% and 42% of the world average by the 1980s and 1990s, respectively [2]. Notwithstanding this dismal agricultural performance record, SSA, in the same period, has seen substantial increases in population figures. The population of the sub-region increased from 221 million in 1960, to 856 million in 2010 and recently crossed the billion mark in 2017 [3,4]. The inadequacy of local production is demonstrated by the quantum and trajectory of food imports, even including staples such as maize *(Zea mays)* and rice *(Oryza sativa)*, into the region in the last 50 years [1]. Based on the statistical database of the FAO, maize is the most important staple in SSA, occupying some 38.7 million hectares—the largest of all staples—and with annual grain production is estimated at about 79 million metric tonnes as of 2018. Maize also contributes the highest per capita calorie consumption in SSA, where more than 208 million people depend on it for food security and economic wellbeing [5]. It is against this backdrop that there are increasing doubts about the capability of the SSA region to achieve food security by 2050 [6].

It is pertinent to note, however, that the geography of SSA, coupled with the prevailing economic situation—poverty still being endemic—put the prices of most imported foods beyond the means of a large proportion of the rural population in the region [1]. Dependence on own food production is, thus, most crucial in the context of resource-poor rural communities where participation in formal agricultural markets is often inefficient. Reliance on own production, however, also implies the need to ensure adequate production to assure food security. Identifying the factors that underpin current yield levels is thus vital to dealing with current poor yields [7]. However, significant heterogeneity in productivity exists, even within the same agro-ecological regions and villages, and thus, the factors would also be heterogeneous [8].

### 1.1. Factors Influencing Yield Levels

A crucial strand of the farm productivity literature on smallholder family farms revolves around yield levels and variability and the factors that impinge on these. Yields have been noted to vary significantly not only between seasons and agro-ecological regions [9,10], but even within the same regions, villages or even adjoining plots [11,12]. At a much broader spatial scale, yield levels and their spatial variability are often ascribed to varying climatic conditions [9,13,14]. This is so given that much of the cultivation in SSA in general and Ghana in particular is still undertaken under rainfed conditions [15]. To underscore the importance of the contribution of climatic factors to yield variability, it is pertinent to note that other factors such topographic indices and soil physical properties still vary with climatic conditions [16,17]. The extent of the influence of climate on crop yield level and its variability is, however, context- and crop-specific [9,18]. While they serve other purposes, yield analysis at coarse spatial scales such as regions and countries are of limited relevance smallholder farmers who constitute the bulk of producers in SSA.

Crop yield analysis at the farm level is of greater relevance to the smallholder farmer. Findings of significant yield differences at such micro scales such as the plot level [2,19,20] suggests that village-wide, biophysical constraints such as rainfall amount and distribution as well as other climatic factors can be discounted as principal sources of the observed variability. Thus, the substantial plot level yield variability, including on adjoining plots [12,21], indicates the contribution of other factors to yield levels. The most-recurring determinants of yield level are soil fertility and fertilization levels [2,21] and access to and use of improved technologies including improved seeds [15,22,23]. Other relevant factors include specific management practices such as timing and density of planting [23–25], timing and

frequency of weed control [26], management of preceding season's crop and other vegetal residue [27], as well as effective control of pests and diseases [28]. Apart from such conventional farm management factors, less conventional ones such as chemical and biological control of weeds are increasingly gaining in popularity [29]. Chemical control of weeds has also been shown to have important implications for maize germination rates and yields [30]. Most of these reviewed studies are, however, constrained by their monodisciplinary approach and/or overemphasis on technical solutions to the neglect of a wider lens that integrates the social, economic and political contexts [31].

*1.2. Role of Socioeconomic Factors*

As Mueller and Binder [32] argued, yield determinants are as much sociopolitical and economic as they are environmental because the former influence farmer decision making with regards to management practices and this, alongside local environmental conditions, determines the biophysical conditions crops experience during development. To illustrate, while poor soil fertility and weed control may be two important management factors influencing yield levels, they are often driven by low purchasing power [33]. That is, limited financial means in the household at crucial times of the farming season implies that farmers are often not in position to purchase the required quantities of fertilisers to augment soil fertility levels or pay for hired labour to control weeds on their farms timeously. There is thus a need to not only broaden the methods and data sources but also the approaches adopted to analysing yields and their variability. The sustainable livelihood approach (SLA) offers a unique perspective to a more nuanced understanding of the dynamic and complex rural setting given its commitment to locally-embedded contexts, place-based analyses and the perspectives of the poor regarding the challenges they are confronted with [34]. The SLA and its associated framework are concerned with understanding how the differential capabilities of rural households influence the outcomes of their livelihood strategies [35], in this case, yields from their farms. The opportunities and constraints that farmers contend with—their socioeconomic milieus—constitute their vulnerability contexts which, in turn, influences the assets—human, natural, financial, social and physical assets—that they control [36,37]. It is these assets that fall within the rubric of socioeconomic factors that influence crop yield levels.

Yield levels can also be understood as an object of choice for farmers, and thus, their analyses should give weight to both biological and economic realities of crop production [38]. Thus, within the same villages, explanations of yield levels should not only require information on the biophysical environment and crop management but also farmer characteristics and the socioeconomic constraints within which farmers operate [39]. To illustrate, in assessing the relative contribution of five main groupings of variables—inputs, soils, landscape, climate and management—to explaining grain yields, Kraaijvanger and Veldkamp [18] found that farmer impacts—management—was the most important, explaining about 45% of the observed yield variability. It is important to note that while smallholders tend to be risk-averse, they are also rational, and as such, take farm management decisions by considering a complex web of socioeconomic factors [40,41]. Despite the central role that farmers play, their perspective as well as their socioeconomic milieu are often not adequately analysed in studies on crop yield level.

It is important to point out that while some existing literature in this field includes the influence of socioeconomic factors [42,43], not many actually incorporate these into their analyses [31]. When considered, socioeconomic factors often include farmer knowledge, access to capital and credits, markets structure and access and institutional factors such as governmental policy and support and extension services [42,44,45]. Other equally important but less often considered socioeconomic factors include the role of land tenure dynamics, distances of plots from homestead, presence and importance of non-farm activities and farmers' management of risks [31]. For example, tenure systems in operation can in diverse ways contribute to land fragmentation and reduced fallows. Yield levels are bound to suffer in contexts where shifting cultivation is predominant, but where these new dynamics do not induce adoption of high-yielding varieties and necessary farm management changes including

improvement in fertilization levels as land use is being intensified [11,46]. Similarly, transitioning from shifting cultivation to more intensified land use calls for the adoption of higher yielding varieties of seeds, which in turn should be accompanied by improved levels of fertilization [47,48]. Furthermore, it is important to understand people's priorities and motivations and not assume that people are always entirely dedicated to maximizing their production or incomes [37].

While the individual factors' effect on current yield levels has been separately well-studied, how the different drivers interact with each other as well as their magnitude of contribution has not garnered the needed attention. It is also clear that those studies conducted at this scale are often limited by the scope of data they rely on and their monodisciplinary focus. These studies either rely on modelling, surveys and/or field experimentations to analyse crop yield levels and their variability, or else use spatial scales that cannot tease out the actual dynamic interactions at the micro scale. *The present paper therefore uses an integrated approach to investigate the sources of current crop yield levels and their variability at the plot levels in two maize farming villages* in the Eastern region of Ghana. The study thus draws theoretical inspiration largely from the sustainable livelihood framework (SLF) as espoused by DfID [37] and to a lesser extent from the induced innovation theory by Hayami and Ruttan [47]. It uses a mixed methods framework to incorporate household and farm survey data and remotely sensed data of plots with qualitative data from photo-elicitation interviews to shed more light on the sources of the present poor yields of maize from the region.

## 2. Materials and Methods

### 2.1. Study Villages

The study was carried out in two villages—Asitey and Akatawia—in the Eastern Region of Ghana. The relatively less rural Asitey (Lat. 6.129601°, Lon. −0.013253°) is less than a kilometre from Odumase, the municipal capital of the Lower Manya Krobo Municipality, while the relatively more rural Akatawia (Lat. 6.283055°, Lon. −0.128794°) is about 9 kilometres from Asesewa, the capital of the Upper Manya Krobo District (Figure 1). In terms of climate, both study sites are located in the Semi-Equatorial climatic belt of West Africa with similar mean annual rainfall values. As such, both locations experience the bimodal type of rainfall with the major rainy season falling between March and early August, while the minor one falls between September and late October. Thus, biannual crop cultivation is often practised with longer and shorter rainy seasons coterminous with the major and minor farming seasons. Apart from the wet season, the region also experiences a dry season, the Harmattan, between November and March. With regards to agro-ecology, both study sites are located in the moist Semi-Deciduous forest zone.

Table 1 summarises the main climatic and population characteristics of the two administrative districts in which the two study villages are located. Asitey is overlooked by the Akwapim-Togo Ranges, and thus, has isolated hills, notable among them being the Krobo Hills, while Akatawia is relatively flat with a slightly undulating landscape. Notwithstanding these differences, agriculture is naturally a major economic activity in both study locations. While maize is, by far, the most important food crop in both villages, Asitey is also noted for the production of mangoes, peppers, pineapples, tomatoes, okra and watermelons, while Akatawia is noted for cassava, plantain, cowpea, pepper and oil palm. Given the proximity of the Volta Lake to the two locations, fishing is a major potential off-farm activity. Livestock rearing and trading in agricultural produce are important off-farm activities at Asitey. In Akatawia, other off-farm economic activities include cassava processing into *gari*, carpentry and beads making, as well as the production local gin (*Akpeteshie*) and palm oil. Again, while both villages have local markets serving them within their respective districts, Asitey has a more direct linkage to the metropolitan cities of Accra and Tema. The choice of both districts and villages is based, among other factors, on the fact that they represent a typical farming community on the agricultural dynamism/potential spectrum in the Ghanaian context [51].

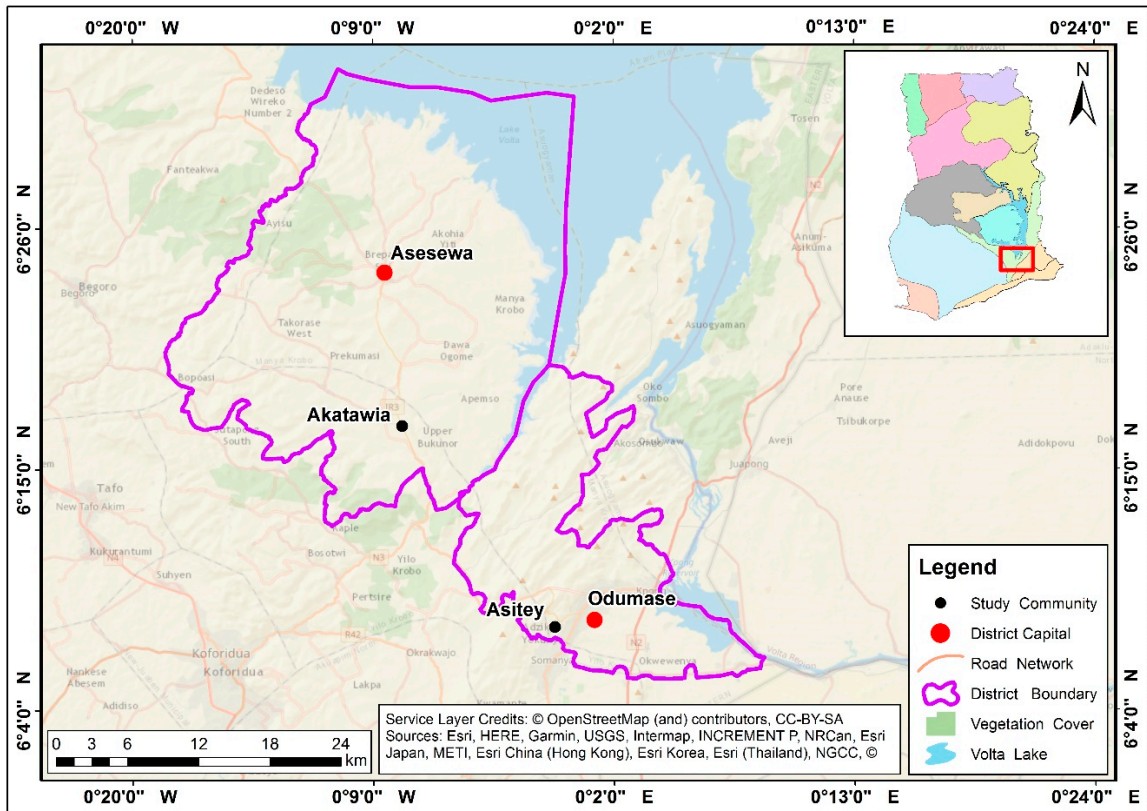

**Figure 1.** Map of the study districts showing the location of Asitey and Akatawia.

**Table 1.** Geographic, climatic and socioeconomic characteristics of the study districts.

| Variable | Lower Manya Krobo (Site: Asitey) | Upper Manya Krobo (Site: Akatawia) |
|---|---|---|
| Altitude (metres above sea level) | 150–600 | 120–440 |
| Annual rainfall range (mm) | 900–1150 | 900–1500 |
| Mean temperature range (°C) | 26–35 | 26–32 |
| Relative humidity (g/m$^3$) | 70–95 | 75–95 |
| Population density (pers. Per sq. km) | 293 | 84 |
| Proportion of rural population (%) | 16 | 84 |
| Proportion of pop engaged in agric | 66 | 73 |
| Major market centres | Somanya, Odumase, and Kpong | Akatim, Sekesua and Asesewa |

Source: Based on data from GSS [49], GSS [50].

*2.2. Plot Selection and Household Sampling*

The paper uses data collected from the two study villages in a cross-sectional comparative mixed-methods framework. The initial quantitative fieldwork commenced in January 2016, while the final qualitative fieldwork was concluded in February 2019. A multi-scale sampling strategy was deployed. The sample for the quantitative survey was drawn from that which was used by the second round of the Africa Intensification (AFRINT II) survey and described in detail by Djurfeldt [52]. The multi-scale sampling approach entailed country, region, district and village selection with the sampling of farm households within individual villages being the last stage. With the exception of the last stage, all others—country, region, district and village—were purposively selected. The selection of region and villages was done on the basis of such factors as agricultural potentials and with a view to capturing the dynamism of agricultural production systems in Ghana. Furthermore, we aimed to study the prevailing above-average regions in terms of their ecological and market endowments,

and thus, excludes most extreme cases at both ends of the agricultural dynamism/potential spectrum. The final sample of 30 households from each village was arrived at through a simple random sampling approach. Given the time span from when the original AFRINT II sample was drawn and the present survey, it was to be expected that some farming households would have dropped out through ageing, death, out-migration or stopped farming altogether. Wherever this was the case, the next-of-kin, closest relative or another member of the same household was drafted to replace the dropout. The main respondents for the household surveys were household heads.

In terms of the selection of farm plots, maize was the primary crop of interest. It was commonplace to find a household operating multiple maize plots, often at multiple locations. Such households were asked to indicate the main plot, which was then selected. Others, however, had a single maize plot operated by the entire membership of the household. Additionally, there needed to be a sufficient level of homogeneity within sampled maize plots for the purposes of the present study. Thus, in a few instances where we discerned significant heterogeneity in terms of slope, cropping history or planting time, even when the farmer regarded a field as a single unit, we further divided such fields into smaller, more homogeneous units. As a result, for the 60 households, a total of 87 comparable maize plots were surveyed. In each of the plots, a 4 m × 4 m subplot was demarcated at the approximate centre. The procedure for the demarcation of the subplot has been detailed by Wahab et al. [53] and Wahab [54]. To eschew bias, the delineation was done prior to maize planting. It is within these subplots that the crop cutting was done to derive crop cut yields of maize.

With regards to the surveys, plot surveys were carried during the 2016 major farming season spanning March to July of that year. Data collected from individual maize fields included weed coverage and height, field erosion status, maize density and height in the 16 m$^2$ subplot. At the start of the farming season, soil sampling was undertaken from the 0–15 cm layer within the same subplot according to the guidelines provided in Yeboah et al. [55]. The soil samples were then analysed to derive soil chemical and physical properties such as soil organic matter content, pH, cation exchange capacity percentage of sand, silt and clay and total nitrogen percentage, among others. In the same 16 m$^2$ subplot, researchers carried out crop cutting as described by FAO [56]. Apart from the plot surveys, household surveys were also carried out at the end of the farming season after all the fields had been harvested. Data collected from the household surveys include household socioeconomic characteristics—household size, labour availability, area of farmland under each household's control, income bracket, proportion of off-farm income and agriculture information sources, among others. There were questions regarding field cropping history in the preceding three years, ownership structure, crop residue management and self-reports of current season's maize yields, among others. Data on farm management characteristics, such as method of plot preparation, timing of planting, fertiliser use—timing and quantity—and weed control as well as number of man-hours used for all farm activities from plot preparation to crop harvest, was also collected.

*2.3. Yield Measurement*

Two distinct approaches to yield estimation were deployed in this study; one was based on farmers' self-reports of the outputs from their plots—SR yields—while the other was based on crop harvests by researchers from the 4 m × 4 m subplot—CC yields. The SR yields were derived from the household surveys conducted after harvest, during which farmers were asked to report overall crop output including green maize harvests as well as payments in kind for labour or plot rental. Farmers were implored not to treat the 4 m × 4 m subplot any differently from the rest of the larger plot and not to harvest any maize from the subplots of each maize plot. As far as we could tell, farmers obliged to our request and treated the subplots the same way they did the larger plots. To ensure they did not harvest any green maize from the subplot, maize stands were counted at about four weeks after planting and compared with crops stands at harvest; no harvests were prematurely done by farmers in the subplot, though farmers were at liberty to harvest both green and dried maize from the larger plot

and report quantities during household surveys. Yield data for both SR and CC yields were computed as follows:

$$Yield_{SR} = \frac{Crop\ output_{SR}}{Area_{GPS}} \tag{1}$$

$$Yield_{CC} = \frac{Weighed\ output_{CC}}{Area_{GPS}} \tag{2}$$

*Yield*$_{SR}$ = Farmers' self-reported maize yields; *Crop output*$_{SR}$ = Total maize output in kg from the whole field as reported by farmers and includes quantity harvested green; *Area*$_{GPS}$ = Field area measured by walking the perimeter of each field using a Garmin 64S GPS device; *Yield*$_{CC}$ = maize yields computed based on crop cuts; *Weighed output*$_{CC}$ = weight of grains in kg harvested from the 4 m × 4 m subplot.

## 2.4. Statistical Analysis

Household and field survey data were collected using the Ona mobile data solution and application (Ona Kenya Ltd, Nairobi, Kenya). The dataset was exported to an Excel spreadsheet and then cleaned. Data outliers were verified with concerned households and rectified where possible or otherwise trimmed at the 99th percentile. Data validation was done through a feedback workshop with the smallholder farmers in both villages. For instance, we had initial concerns regarding the yield levels, particularly in Asitey, but these were eased by the farmers who ascribed the low yields to the devastating effects of the outbreak of the fall armyworm (*Spodoptera frugiperda*) during the 2016 maize farming season.

The dataset was imported into IBM SPSS Statistics v.25 for analyses. Initial sample characteristics tests conducted included tests of normality and for significant differences in mean yields between study villages. A Shapiro-Wilk's test ($p < 0.05$), coupled with a visual inspection of respective histograms, normal Q-Q plots and box plots for both study villages shows that both crop cut and self-reported yields deviate from a normal distribution [57]. It is important to note, however, that other factors such as sample size can equally yield significant results even for a dataset that is otherwise normally distributed [58]. Thus, a non-parametric, a Man-Whitney U test to compare the mean SR and CC yields ($p = 0.006$ and $0.008$) implies that there are significant differences between Asitey and Akatawia in the mean ranking of the yields [58].

Subsequent to these initial exploratory analyses, a principal component analysis (PCA) was performed to extract the key soil variables and to visualise patterns in the dataset. Finally, to test for the magnitude of contribution of the various factors to maize yields, multiple linear regression (MLR) was performed on the dataset. The MLR used either the SR or CC yields as the dependent variable and soil, management and socioeconomic variables as independent variables. In total, there were six models. For each study village, three models were run one each for SR, CC yields and the pooled dataset. This was done because the factors that drive maize yields were expected to differ from village to village. The coefficient used for the interpretation of the strength of the influence of individual variables is the standardised coefficient (β), which allows for comparison both within and across factor categories. Multicollinearity was absent given that none of the correlation coefficient between the variables was 0.70 or more. Indeed, the highest coefficient was 0.59.

## 2.5. Principal Components

Principal component analysis (PCA) for the soil cluster of factors enabled us remove collinearity and reduce the dimensionality of the dataset from 22 soil variables to 12. Sampling size was found to be adequate for the PCA given the Kaiser-Meyer-Olkin value 0.65 [59]. Bartlett's test of sphericity was significant—less than 0.01 [58]. The PCA returned four components with Eigenvalues > 1. Together, all four components account for 71% of the variance; distributed as 29%, 17%, 13% and 12% of the total variance being accounted for by the first, second, third and fourth components, respectively. As Table 2 shows, the percentage of silt in the soil has the highest communality, followed by the cation exchange

capacity, magnesium, pH, nitrogen and percentage of sand in the soil, in that order. Of the retained soil variables, sodium has the lowest communality.

**Table 2.** Rotated component matrix of principal component analysis of the soil variables.

| | Communalities | 1 | 2 | 3 | 4 |
|---|---|---|---|---|---|
| Cation exchange capacity | 0.843 | 0.896 | 0.135 | 0.020 | 0.147 |
| Nitrogen percentage | 0.781 | 0.881 | 0.037 | 0.046 | 0.032 |
| Magnesium | 0.810 | 0.880 | 0.166 | 0.090 | 0.000 |
| Percentage of clay | 0.660 | 0.714 | −0.043 | 0.360 | 0.140 |
| Potassium | 0.642 | 0.537 | 0.267 | 0.140 | 0.512 |
| pH | 0.805 | 0.118 | 0.857 | 0.238 | 0.024 |
| Percentage of sand | 0.729 | −0.121 | 0.789 | −0.303 | 0.016 |
| Sodium | 0.496 | 0.218 | 0.668 | 0.032 | 0.020 |
| Percentage of silt | 0.927 | 0.065 | 0.071 | 0.958 | −0.034 |
| Soil electrical conductivity | 0.503 | 0.452 | −0.081 | 0.537 | −0.051 |
| Average soil penetrability | 0.668 | 0.011 | −0.270 | 0.053 | 0.769 |
| Phosphorus | 0.503 | 0.113 | 0.286 | −0.178 | 0.736 |

Extraction Method: Principal Component Analysis. Rotation Method: Varimax with Kaiser Normalization. Rotation converged in five iterations.

Table 2 also shows the rotated component matrix with soil factor 1 being loaded heavily by CAC, nitrogen percentage, magnesium, percentage of clay and potassium. Factor 2 is strongly loaded by soil pH, percentage of sand and sodium, while factor 3 is strongly loaded by percentage of silt and, to a lesser extent, soil electrical conductivity. The final factor is also strongly loaded by average soil penetrability and phosphorus. The PCA thus creates an index of four main soil indices for the regression analyses.

## 2.6. Aerial Photography

Aerial photography of the plots was carried out using an unmanned aerial vehicle (UAV) system. Plots were flown one–three times in the course of the farming season between when the crops were ~4 and ~12 weeks old. The UAV system comprises an Enduro quadcopter (Agribotix, CO, USA), which is powered by a Pixhawk flight control system (3D Robotics, Boulder, Colorado, CA, USA) and mounted with two GoPro Hero 4 cameras (GoPro Inc., San Mateo, CA, USA). The cameras are identical, except that one is modified such that the red band is replaced by a near infrared (NIR) band. During missions, the UAV system is flown autonomously at a height of ~100 m above ground in a survey grid format at a speed of 14 m/s. The flying altitude of ~100 m was considered an ideal balance between flying clear of the tallest trees in the landscape but low enough to capture the crop canopy in sufficient detail. Captured images are processed in post-flight processes including geotagging, mosaicking and georeferencing. The resultant mosaic has a spatial resolution of 3 cm.

Using the Map Algebra tool in ArcMap, the main component of Esri's ArcGIS (Esri, Redlands, CA, USA) suite of geospatial processing software, and after projection, transformation and clipping of individual bands of each plot mosaic, the green normalised difference vegetation index (gNDVI) was extracted. This was done by finding the ratio of the near-infrared and the green bands of the processed mosaic from the modified camera:

$$gNDVI = \frac{(NIR - G)}{(NIR + G)} \tag{3}$$

where *gNDVI* = green normalised difference vegetation index; *NIR* = reflectance captured in the band 1 of the modified camera; *G* = reflectance captured in the band 2 of the modified camera. The variances between these two bands enable the assessment of relative crop density and vigour within the plots [53]. The resultant crop health maps were then printed and returned to the study sites for co-interpretation in conjunction with plot operators in photo-elicitation interviews.

*2.7. Photo-Elicitation Interviews*

Overall, 24 photo-elicitation interviews were conducted with smallholder maize farmers—12 from each study village. Each of the 24 interviews lasted between 30 and 45 minutes. The 12 interviews in each study village comprised four with operators of averagely-performing plots, four with operators of well-performing fields and the final four with operators of poorly-performing fields in terms of yields. The holistic view offered by the UAV imagery of the plots afforded farmers an atypical view of their plots from a vantage point. This served to elicit more insightful perspectives from operators of the plots. Apart from validating sections of plots with varying crop vigour as depicted by the aerial images, farmers were asked to assign reasons for poor and healthy patches and proffer explanations for the spatial locations, distribution and characteristics of these.

## 3. Results

*3.1. Household Characteristics and Farming Systems Practices*

With regards to household characteristics, both study villages exhibit relatively large household sizes—averaging about seven members per household. Of this, close to half—2.6 in Asitey and 2.3 in Akatawia—is aged 16 or younger (Table 3). Given the average age of the household head of about 55 years, we found that the demographic profile of the households is ageing, and this can have important implications for the active labour force needed on the farm. In terms of plot characteristics, average maize plot sizes are relatively small; about one acre in both villages. This finding is instructive given the relatively large household landholding under fallow; about 2 ha in Asitey and close to 3 ha in Akatawia. This notwithstanding, maize occupies the largest share of cropland for a single crop. The relatively small maize plot sizes in spite of apparent availability of land may be attributed to two immediate factors—distances of plots and the tenure system in operation. While about a quarter of the plots in both study villages are within a 100-meter radius of the homesteads of operators, a significant majority of farmers travel long distances on each farming day—2.5 km in Asitey and 1.2 km in Akatawia.

**Table 3.** Descriptive characteristics of households and plots from both study villages.

| Descriptive Characteristics of Households and Maize Plot | Study Village | |
|---|---|---|
| | Asitey | Asitey |
| Average household size | 6.9 (0.5) | 7.0 (0.5) |
| Average economically-inactive membership (i.e., <age 16) | 2.6 (0.3) | 2.3 (0.3) |
| Average maize plot size, ha | 0.41 (0.05) | 0.42 (0.04) |
| Average land under fallow, ha | 1.7 (0.4) | 2.9 (0.4) |
| Average maize yields, t/ha | - | - |
| Crop cut yields, t/ha | 2.36 (0.16) | 2.68 (0.10) |
| Self-reported yields, t/ha | 0.98 (0.06) | 0.93 (0.08) |
| Proportion using herbicides (%) | 55 | 80 |
| Proportion using improved seeds (%) | 9.52 | 6.67 |
| Proportion applying inorganic fertiliser (%) | 43 | 80 |
| Inorganic fertilization rate, kg/ha | 15.5 (24.0) | 27.0 (29.5) |
| Weeding frequency | - | - |
| Never weeded (%) | 1 | 0 |
| Once (%) | 62 | 58 |
| Twice (%) | 36 | 40 |
| Three or more times (%) | 1 | 2 |
| Average distance of plots from residences (km) | 2.5 | 1.2 |

Source: Field survey, 2016. NB: values in parenthesis are standard errors of the mean.

The land tenure system in operation in both Asitey and Akatawia is predominantly private in nature. Four main tenure arrangements obtain in the two villages as shown in the Figure 2. In Akatawia,

almost two-thirds (62%) of the overall survey sample cultivated maize on inherited, and thus, own-plots. Almost a third (31%) of the sample rented their plots with only a small proportion outrightly purchasing their plots. In Asitey, however, almost half (48%) of the plots were rented, with nearly a quarter (24%) of the plots having been inherited. An important distinguishing characteristic of plot ownership in Asitey is that almost a third of the sample fields (29%) is accessed under informal arrangements.

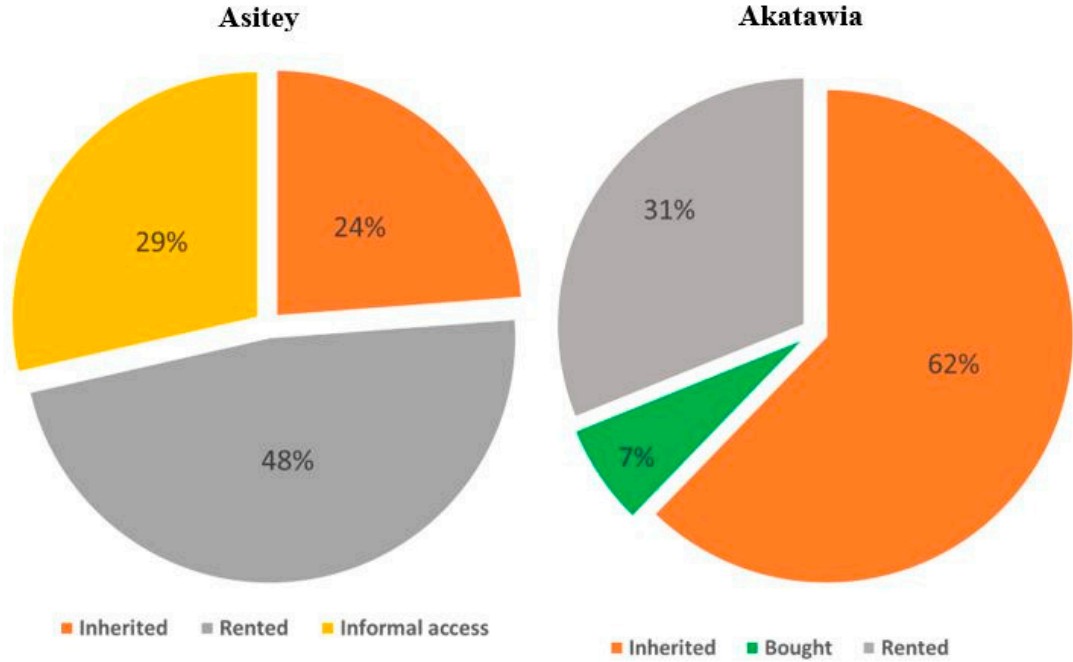

**Figure 2.** Maize farm plot ownership structure in both villages.

In terms of farming system characteristics, plot preparation for the major season usually begins in February, well before the start of the rains in March–April. Plot preparation is either manually done using the cutlass and hoe or mechanically done using the tractor. However, the few fields in Asitey that use mechanical methods of preparation tend to be ploughed along old plough lines, which are along, rather than across, slopes. For fallowed land, preparations for maize cultivation normally start with slashing and burning the cut and dried vegetation. An emerging practice that has now become common is the application of glyphosate-based herbicides as part of the plot preparation process. Even for fallowed plots, this is done two to four weeks after burning before planting is done. Hitherto, planting was done a few days after burning or without burning; in the latter case, the decaying vegetation and/or previous season's crop residue could act as mulch for the maize plants.

For farmers who have been able to prepare their plots in time, maize planting ideally starts within a few days of the first major rains. Planting is done either in rows using ropes and sticks or randomly using the cutlass or stick as the main planting tools. Within our 16 m$^2$ subplot, maize planting density ranged from an average of 3.63 plants m$^{-2}$ in Akatawia to 3.75 plants m$^{-2}$ in Asitey. This is comparable to an average of 46 and 53 ears from the same 16 m$^2$ subplots at harvest. In terms of seed type, only a small proportion—10% and 7% of the plots were planted with high-yielding varieties of maize seeds with the vast majority, 79% and 76%, in Asitey and Akatawia, respectively, being planted with recycled seeds. Most of the plots—71% and 69% in Asitey and Akatawia, respectively—were monocropped, with a quarter of the plots—26% in Asitey and close to a third, 29%, in Akatawia—planted with one intercrop. The intercrop of choice in both study villages is cassava.

In terms of plot management, key among the practices include weed control and fertilization. A substantial proportion of the plots—58% in Akatawia and 62% in Asitey—were weeded once, with a further 40% and 36% in Akatawia and Asitey, respectively, weeded twice. The chemical method of weed control is the most-common; applied on 80% of plots in Akatawia and 55% of plots in Asitey, with the

remaining proportion being weed-controlled mechanically using the cutlass and hoe. The commonest herbicides in use in our study villages are paraquat, atrazine, glyphosate and 2,4-D, with a substantial proportion of plots being sprayed with a mixture of two or more of these in a bid to fight off the FAW infestation. However, we found during the field surveys some crop stands suffering from significant burns from the herbicides for which the farmers admitted the nozzles of the knapsack sprayer may not have been properly positioned. While none of the plots were fertilised with organic fertilisers, a substantial proportion—80% in Akatawia and 43% in Asitey—applied inorganic fertiliser. However, application rates for the plots that applied some amount of fertiliser was meagre—16 kg/ha in Asitey and 27 kg/ha in Akatawia. The most common inorganic fertiliser used in our study villages is NPK 15:15:15, usually applied when crops are about 5 weeks old, with a few farmers top-dressing with urea.

### 3.2. Description of Variables

Table 4 presents the results of the descriptive statistics of the variables used for the MLR. The values for mean crop cut yields were slightly higher in Akatawia—2.6 t/ha—than in Asitey—2.3 t/ha—though the Asitey values had a wider range than those of Akatawia. Farmers' self-reported yields were significantly lower than the CC yields. Between the study villages, mean SR yields in Akatawia were slightly lower—0.92 t/ha—than those of Asitey—0.99 t/ha. With regards to timing of planting relative to first major rains, on average, Akatawia plots were planted later—5.2 weeks—compared to Asitey plots, which on average were planted with maize after 4.6 weeks after the first major rains. While mean weed height at eight weeks after planting were comparable in both villages—26 cm and 23 cm for Asitey and Akatawia, respectively; the difference between the two study villages increases to 35 cm and 25 cm, respectively. Also interesting were the results relating to inorganic fertiliser use rate, which we find were rather low, averaging 16 kg/ha in Asitey and 27 kg/ha in Akatawia. These low fertiliser application rates are not surprising given the current institutional context whereby the fertiliser, which used to be given virtually free of charge to farmers a few years ago, are now being sold, albeit at 50% discount of their market prices.

**Table 4.** Description of variables used for the Multiple Linear Regression.

| Variables | Asitey (*n* = 42) | | | Akatawia (*n* = 45) | | |
|---|---|---|---|---|---|---|
| **Dependent Variables** | **Range** | **Mean** | **SD** | **Range** | **Mean** | **SD** |
| Crop cut yields (CC yields) kg/ha | 437–5274 | 2363.00 | 1053.06 | 924–4458 | 2676.00 | 636.30 |
| Self-reported yields (SR yields) kg/ha | 395–3798 | 989.00 | 347.65 | 390–4745 | 923.00 | 445.28 |
| *Independent variables* | | | | | | |
| Soil PC 1 [nitrogen %, magnesium, % of clay etc.] | −1.73–1.90 | −0.12 | 0.96 | −4.62–2.36 | 0.11 | 1.04 |
| Soil PC 2 [soil pH, % of sand, % sodium] | −1.20–2.66 | 0.14 | 0.84 | −5.05–2.13 | −0.13 | 1.12 |
| Soil PC 3 [% of silt and soil electrical conductivity] | −1.64–4.07 | 0.29 | 1.12 | −2.37–2.63 | −0.27 | 0.79 |
| Soil PC 4 [average soil penetrability and phosphorus] | −1.80–3.61 | 0.31 | 0.92 | −2.31–3.34 | −0.29 | 0.99 |
| Timing of planting relative to first rains (weeks) | 2–7 | 4.76 | 1.08 | 2–7 | 5.20 | 0.99 |
| Inorganic fertiliser application rate [kg/acre] | 10–225 | 37.49 | 59.34 | 15–300 | 67.78 | 72.20 |
| Maize planting density [16 m$^2$ subplot] | 32–96 | 59.74 | 16.07 | 35–90 | 57.98 | 13.65 |
| Mean weed height at 4 weeks after planting (cm) | 0–59.72 | 25.56 | 14.78 | 0–57.55 | 22.79 | 16.13 |
| Mean weed height at 8 weeks after planting (cm) | 0–86.96 | 35.47 | 20.90 | 0–56.70 | 25.27 | 18.76 |
| Household income level | 1–2 | 1.21 | 0.42 | 1–3 | 1.18 | 0.39 |
| Plot ownership | 1–4 | 2.81 | 1.11 | 1–4 | 1.69 | 0.92 |
| Proportion of non-farm income (%) | 0–100 | 43.10 | 26.69 | 0–80 | 35.00 | 24.47 |
| Total family labour used [man-hours] | 0–401 | 99.28 | 80.90 | 0–403 | 105.51 | 91.75 |
| Total hired labour used [man-hours] | 0–264 | 44.02 | 68.04 | 0–161 | 35.66 | 41.77 |
| Total voluntary labour used [man-hours] | 0–74 | 6.31 | 14.23 | 0–147 | 10.31 | 30.75 |

Note: Household income level coded categorically (in USD) as: 1 = 0–100 $, 2 = 101–200 $, 3 = 201–300 $; plot ownership structure as 1 = inherited, 2 = bought, 3 = rented, and 4 = informal access.

### 3.3. Factors Affecting Maize Yields

From the multiple linear regression runs, the factors determining yields are not consistent across the study villages and yield measures. The results of the MLR are presented in Figure 3a–d as well as Table 4 for both pooled and individual villages. As expected, there were negative correlations between yields and timing of planting; these were somewhat consistent across all models. Also expected were the positive linear relationships between both yields and quantity of fertiliser applied and plant density in the 16 m$^2$ subplot (Table 4). The factors with inconsistent regression results include soil principal component 4, average weed height at both after four and eight weeks after planting, household income levels and voluntary labour used on the field.

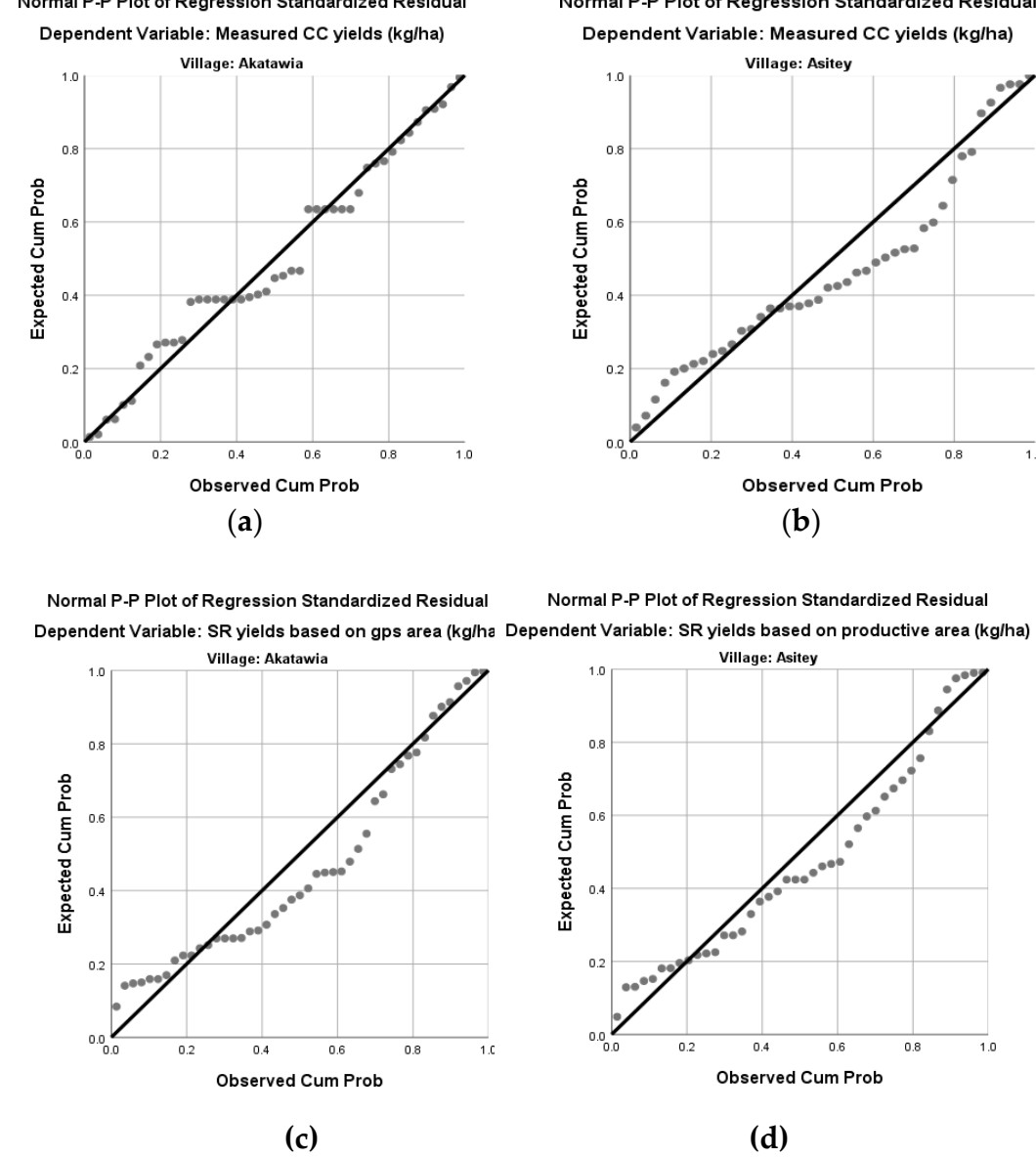

**Figure 3.** Normal probability-probability plot of regression residuals showing deviations CC and SR yields at both Akatawia and Asitey: (**a**) normal p-p plot of regression residuals for Akatawia using crop cut yields as dependent variables; (**b**) for Asitey using crop cut yields as dependent variables; (**c**) for Akatawia using farmers' self-reported yields as dependent variables; (**d**) for Asitey using farmers' self-reported yields as dependent variable.

When pooled, the models do not have adequate explanatory power (Table 5). Table 5 reports on the factors influencing crop yields in both study villages. Table 5 reports on six regression models; two for Asitey—one each for CC and SR yields—and the same for Akatawia with the last two models relying on the pooled data. As a general observation, model three, which relies on the CC yields of Akatawia, was explained the most—$R^2$ = 0.32. With regards to the individual factors, as expected, all models, bar model 4, showed a significant and inverse relationship between timing of planting relative to the advent of first major rains and yields. For model 3, the MLR predicts timing of planting relative to onset of rains, soil PC 3 percentage of silt and soil electrical conductivity), maize planting density and household income level as significant factors. A significant regression equation was found (F (2,42) = 9.782, $p$ < 0.000), with an $R^2$ of 0.32. Predicted CC yields is equal to 4563.65–337.5 (MAIZE PLANTING TIME)—426.94 (INCOME GROUP), where maize planting time is measured in weeks after the first major rains and coded as 1 = 1 week after rains, 2 = 2 weeks after rains, 3 = 3 weeks after rains, etc. and income group coded as 1 = 0–100 $, 2 = 101–200 $, 3 = 201–300 $. Thus, CC yields in Akatawia reduced 337.59 kg for every week's delay in planting and $100 reduction in household income. Other important individual factors include soil component 4—average soil penetrability and phosphorus on the one hand and yields on the other hand, which are positive and significant for models 1 and 5. This means that increased soil penetrability and phosphorus content significantly improved crop yields. For model 3, percentage of silt in soil and soil electrical conductivity has significant and inverse relationship with yields. Also expected was the significant and positive relationship between fertiliser application rate and yields for models 2, 4 and 6. Similarly important is the result relating to the use of voluntary labour and yields for model 2. This implies that access to voluntary labour significantly boosts crop yields.

There were two main factors that had obfuscating results—average weed height at eight weeks after planting and household income levels. While household incomes levels have a positive and significant relationship with the yields for models 2 and 6, it is negative and significant in model 3. This might be attributed to the notoriously unreliable income data reporting. More disturbing was the unexpected positive relationship between yields and height of weeds after planting from models 2 and 6. Perhaps a more accurate variable ought to have been the average weed coverage instead of weed height.

Overall, while none of the variables adequately explain current yield levels, the MLR points to the outstanding importance of the timing of planting. As Figure 4 shows, generally, average CC yields fall as planting delays from the start of the rainy season. Timing of planting thus makes the greatest statistically significant contribution to the prediction of yields—β = 0.53, $p$ < 0.001. This implies that timing of planting uniquely explains 29% of the CC yields in Akatawia. This suggests that building a more complete understanding of current yield levels requires a more nuanced understanding of farmers' motivation and reasoning, which influence their timing of important management activity as planting. Thus, understanding the socioeconomic factors influencing management activities such as timing of planting and level of fertiliser usage becomes critical.

**Table 5.** Yield determinants from the multiple linear regression models of soil, management, and socioeconomic factors for Asitey, Akatawia and the pooled dataset. Dependent variables are the crop cut measured yields and farmers' self-reported yields.

| | Factor Categories | Asitey (n = 42) | | Akatawia (n = 45) | | Pooled (n = 87) |
|---|---|---|---|---|---|---|
| | CC Yields = 2363 kg/ha | SR Yields = 989 kg/ha | CC Yields = 2676 kg/ha | SR Yields = 923 kg/ha | CC Yields = 2514 kg/ha | SR Yields = 955 kg/ha |
| **Soil factors** | | | | | | |
| Soil PC 1 [CAC, nitrogen %, magnesium, % of clay, and potassium | 0.17 | 0.06 | 0.10 | −0.12 | 0.15 * | −0.01 |
| Soil PC 2 [soil pH, % of sand, % sodium] | −0.13 | 0.10 | 0.00 | −0.04 | −0.08 | 0.05 |
| Soil PC 3 [% of silt and soil electrical conductivity] | 0.08 | −0.17 | −0.22 * | 0.00 | −0.05 | −0.04 |
| Soil PC 4 [average soil penetrability and phosphorus] | 0.33 ** | −0.17 | 0.08 | −0.08 | 0.16 * | −0.11 |
| **Management factors** | | | | | | |
| Timing of planting relative to first rains | −0.19 | −0.13 | −0.50 *** | 0.12 | −0.26 *** | −0.15 * |
| Inorganic fertiliser application rate | 0.06 | 0.17 ** | −0.01 | 0.36 *** | 0.06 | 0.12 * |
| Maize planting density [16 m$^2$ subplot] | 0.04 | 0.21 * | 0.22 * | 0.09 | 0.09 | 0.15 * |
| Average weed height at 4 weeks after planting | 0.16 | 0.04 | −0.18 | −0.13 | 0.00 | −0.05 |
| Average weed height at 8 weeks after planting | −0.07 | 0.3 ** | −0.07 | 0.04 | −0.1 | 0.26 ** |
| **Socioeconomic factors** | | | | | | |
| Household income level | 0.01 | 0.21 * | −0.21 * | 0.14 | −0.08 | 0.22 ** |
| Plot ownership | 0.05 | 0.08 | −0.12 | −0.07 | −0.08 | 0.10 |
| Proportion of non-farm income | −0.19 | 0.01 | −0.03 | 0.01 | −0.15 | 0.1 |
| Total family labour used [man-hours] | −0.13 | 0.06 | −0.05 | 0.05 | −0.08 | 0.01 |
| Total hired labour used [man-hours] | −0.02 | 0.2 | −0.03 | 0.09 | −0.04 | 0.12 |
| Total voluntary labour used [man-hours] | 0.04 | 0.35 ** | −0.02 | −0.18 | 0.02 | −0.04 |
| $R^2$ | 0.11 (model 1) | 0.12 (model 2) | 0.25 (model 3) | 0.13 (model 4) | 0.06 (model 5) | 0.07 (model 6) |
| $R^2$ | - | - | 0.32 (model 3) | - | - | - |

NB: Estimates of factors are given with their statistical significance given as follows: $p < 0.011 \geq$ ***, $p < 0.05 \geq$ ** and $p < 0.10 \geq$ *. Missing values are excluded pairwise, standard residual between −2.206 and 2.545, while Cook's statistic is within a minimum of zero and a maximum of 0.591.

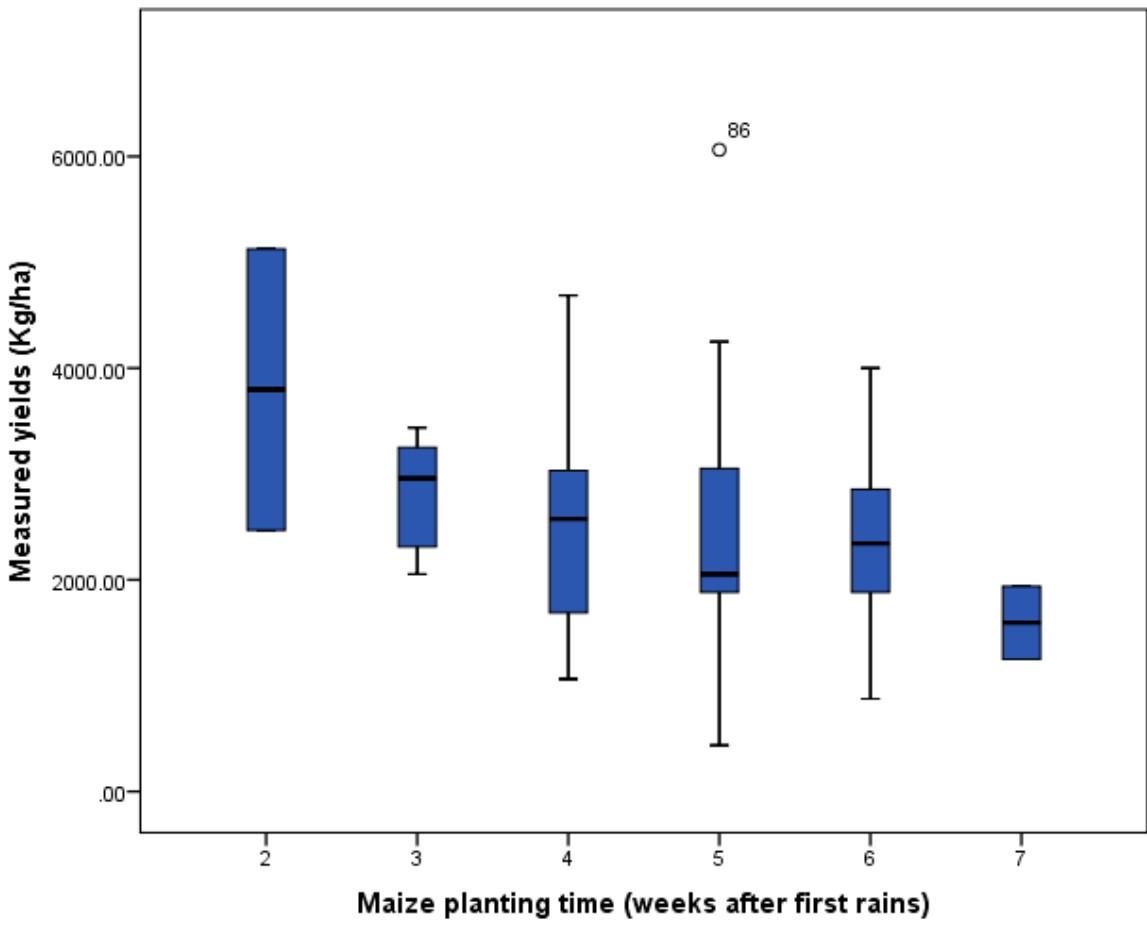

**Figure 4.** Boxplot showing the relationship between timing of maize planting in weeks after the first rains and CC yields.

### 3.4. Spatial Variability in Maize Yields

As shown in Figure 5, a manifest spatial characteristic of the aerial photographs of the maize fields is the generally poor crop vigour on the edges and borders of farms in both Asitey and Akatawia. Figure 5 shows a sample of plots from both locations; plot A is located in Asitey and plot B in Akatawia. Both fields have similar profiles in terms of planting history, cropping system, size and management, except that field A was ploughed prior to maize planting while plot B was not. Additionally, both fields have been continuously monocropped with maize over the last three years. Though both fields are rented, field A was rented under an output sharing arrangement whereby the plot operator pays a third of the crop output to the landowner while field B operator paid a cash lump sum of 70 USD to landowner to cover rental for three years. While agrochemicals were used as part of the plot preparation on A, only mechanical methods—using the cutlass and hoe—were used during preparation for planting on B. In terms of the timing of planting, both fields were planted within a week of each other. Both plots were also fertilised with inorganic fertilisers, twice on A and once on B. While both plots show substantial heterogeneity in within-plot maize vigour, a more detailed examination shows differing spatial patterns in the distribution of poor patches within each plot. A comparison between the two suggests that overall, crops generally have more vigour on plot B than on plot A. More importantly, the poor patches of crops are spatially largely concentrated at the boundaries of plot B—which gives the negative values denoting more stressed crops—compared to A, in which they are more haphazard and more pronounced on the south-west boundary with fallowed field.

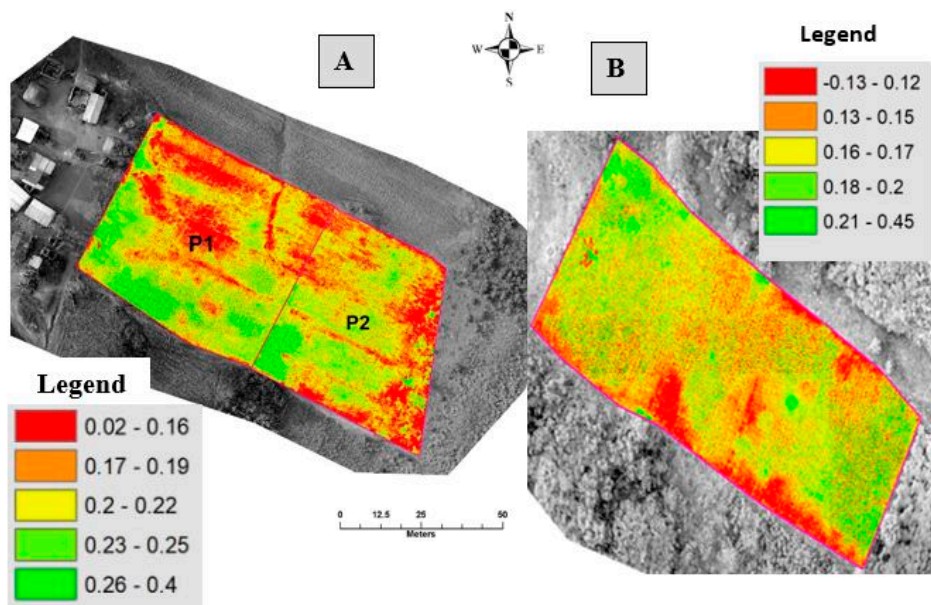

**Figure 5.** Significant within-plot variability in crop vigour at both study locations. NB: Sample plots **A** (in Asitey) and **B** (in Akatawia) were surveyed at 7 and 6 weeks after sowing, respectively. The density and intensity of vegetation present is used to assess vegetation vigour so that the healthier the vegetation, the greener the pixels and vice versa. Thus, the spectrum ranges from 0 to ~0.16 μm for bare lands/soils, and ~0.17 through ~0.45 μm for various levels of vegetation vigour. Based on theory and ground-truthing, maize crops typically fall between ~0.20 and ~0.30 μm (light green pigment) with those above this threshold corresponding to tree canopies and secondary vegetations.

Similar spatial patterns in the within-plot variability in crop vigour can also be discerned on other fields from both villages, with a few exceptions. A common characteristic of the more intensively cultivated plots is that they are leased for a period of up to five years. As a result, farmers tend to want to fully utilise the land to cover rental costs through intensive cultivation. On such plots, poor patches are especially pronounced on the edges, particularly where the terrain is higher than the rest of the plot. Generally, differential spatial and temporal patterns would suggest different types of yield controls [7]. However, using aerial photographs to explain spatial variability in crop vigour based on the cross-sectional data is unreliable [60]. From the interview data, however, farmers attribute two main factors to the absence of crops or crops with reduced vigour on the borders of their plots: picking of seeds by certain species of birds and rodents, and the effects of shades from tree canopies and thick vegetation from adjacent plots. Asked to explain the poor patches, a farmer averred that:

*"Competition from nearby bushes and forest that were not cleared for the current season contributes to the poor patches. This cannot be remedied unless I am lucky and my neighbour also cultivates his field. Partridges also prefer to pick off seeds near the borders of fields and this often leads to sparse maize density on these areas"* —Mr. JT, male, 57, Akatawia.

Similarly, another farmer in Asitey points out that:

*"You could see the poorer patches are generally closer to the borders where my neighbour did not clear his plot. So, I think the lack of breathing space for crops in these areas as well as draining of nutrients by nearby bushes contributed to the poor patches. It is always this way unless you are lucky and your neighbours also cultivate their fields or you are able to clear shades of trees from nearby fields. This can be difficult if your neighbours are also trying to fallow their lands. But if you are lucky and all your neighbours cultivate their fields, you would have a more even crop performance"* —Mr. KYT, male, 62, Asitey.

Finally, yet another farmer in Akatawia shares the following insights on the subject:

*"I see the borders having poor patches, but it is because the neighbouring farmers did not cultivate their own fields. As a result, the shades from other fields affect my own field. So even though there are advantages to farming further away and deeper into the forest, in terms of weed infestation, there are disadvantages too with regards to shades from neighbouring uncultivated lands which affect the vigour of crops on the borders"* —Mr. TAT, male, 77, Akatawia.

The non-cultivation of neighbouring plots is more common in Akatawia than it is in Asitey. In the latter, more fertile farmlands further away from the community have been acquired by the state and managed by the state's Forestry Commission and are thus not rentable per se. There is, thus, relatively more land availability in Akatawia than Asitey. Farmlands, owned by community members, are available and rentable in Akatawia at locations further away from the community centre at relatively low rates equivalent to about 16 USD per an acre per annum. In Asitey, only lands around the village are owned by the community members and are thus rentable. Thus, while poor patches along the borders of plots in Akatawia may be due to adjacent plots laying idle, their presence in Asitey is largely due to shading and competition from mango plantations and cassava farms, and to a lesser extent, fallowed plots. In a similar vein, less plant density discernible along the borders is attributed to the activities of certain bird species picking off seeds within days after sowing. Farmers report that even though they prefer to re-plant where seeds had been picked, sometimes measures such as erecting mannequins become ineffective if the fields are too far from homesteads and farmers are unable to be present on their farms for entire days to drive off birds, at least, until the seeds germinate. Hence, the assertion by Mr. TAT that the advantages of cultivating younger plots further away—more fertile and easier and more effective weed control—are counterbalanced by shading from neighbouring uncultivated lands and the picking of seeds by birds, contributing to sparse crop vigour on the borders.

Furthermore, while care was taken to ensure a certain degree of uniformity in terms of the terrain, the latter was still a contributory factor to within-plot variability in crop vigour. The challenge with topography, especially in Asitey, is succinctly expressed by one farmer: *"The problem with this area* [community] *is that you never get uniform, flat lands. The plot is either too stony or hilly. This is the reason why one can never get a plot with uniform crop performance"* (Mr. TTH, male, 68, Asitey). Similarly, another farmer, in seeking to explain the apparent disparity in crop performance on his fieldposts that:

*"The higher parts have poorer crops because there is inadequate water [moisture] in the soil. The crops in the low lying parts look better not only because there is [relatively] more water [moisture] in the soil but also because the top rich soil, including any fertiliser applied, could be washed off the higher terrain and deposited in the lower-lying parts"* —Mr. PNK, male, 58, Asitey.

Mr. PNK alludes to fertiliser being washed off because rather than cover fertiliser application with dirt, farmers in these communities apply ammonium sulphate crystals and urea at the base of plants. This is done with the view that the application would eventually dissolve into the soil. Thus, on plots with significantly differing topography, higher areas lose the top rich soils and fertiliser applications to low lying areas during torrential rains.

## 4. Discussions

Comparing maize yield levels in the two study villages, there appear to be inconsistences in the two yield measures. On the one hand, average yield estimates based on farmers' self-reported output are slightly higher in Asitey—988 kg/ha—than in Akatawia—923 kg/ha. These were, however, far lower than average yields estimates based on crop cuts for both study villages. Average yields estimates based on the CC measure, on the other hand, were significantly lower in Asitey—2363 kg/ha—relative to those in Akatawia—2676 kg/ha. A discussion of this inconsistency has been extensively done in Wahab [54]. It is important to note, however, that average CC yields are much closer to the preceding three-year district averages of maize yields which ranges from 2280 kg/ha to 1920 kg/ha for Lower

Manya Krobo and Upper Manya Krobo districts, respectively, and also compares well with the national average of 2 tons/ha [61]. Given that the same report estimates attainable yields—maximum yields achievable by resource-endowed farmers on their most productive fields [62]—at 5500 kg/ha, there is significant scope for yield improvement.

Given the much-touted importance of fertiliser application in improving yield levels [2,21], the low application rate of less than 30 kg/ha compared to the recommended rates of about 120 kg/ha for SSA [23] is quite puzzling. Even more disturbing is the claim by some of the interviewed farmers that they perceived no significant differences between the yields of users and non-users of inorganic fertilisers. This notwithstanding, the general opinion as far as fertiliser use is concerned was that, resource availability allowing, most of the farmers would apply fertilisers on their farms. The extant literature also points to limited means of farmers as the main factor for the low application of inorganic fertilisers in SSA [27,63,64]. While the affordability factor may still be important, farmers failure to perceive any significant difference between users and non-users of fertilisers may partly be attributed to non-responsiveness of local soils and seeds. The present study, for instance, found that local, recycled seeds predominate the maize seeds planted—76% in Akatawia and 79% in Asitey—while less than 10% of the fields in either village is planted with improved seeds. The latter has been shown to be significantly more responsive to fertiliser application compared to recycled seeds [47].

### 4.1. Importance of Maize Planting Time

The factors explaining current yield levels are varied, inconsistent and inadequate across both study villages and yield measures. The statistical analyses of the composite of all the factors have shown that timing of planting is an important explanatory factor of CC yields in both study villages. This is in line with the findings of Dobor [25], who recommend changes to earlier spring planting of maize in Hungary based on crop modelling. While delayed planting has been associated with lower yields in South-West Niger, very early sowing immediately after the first rains does not maximise yields either [65]. The importance of the timing of maize planting has been underscored by Adu et al. [24], who averred that under normal rainfall regimes, early planting of maize is associated with higher yields because early planting affords the plants the opportunity to utilise the entire growing season and consequently maximise yields. They further argue that even when rainfall is less reliable at the start of the season, early planting is recommended, as maize plants can tolerate dry spells better in the few weeks of growth than at later stages. It is for this reason that, among others, sowing dates have been recommended as an adaptation strategy against the effects of climate change on agriculture [66]. Farmers are aware of the existence of the so-called ideal planting window, and thus, rarely purposely stagger planting. Indeed, the majority (90%) of the farmers interviewed prefer to plant and do plant entire maize fields within a day or two. Most farmers in this region thus wait for the first rain events within the usual planting period to commence planting [67]. They perceive that the right window to sow maize for the major farming season is between the last week of March to the third week of April. There was agreement among farmers that if sowing was done within this period, crops are likely to receive enough rain at crucial stages of their development and thus do relatively well. Given this insight, farmers thus aim at completing planting within a week. The excerpt below echoes the sentiments shared by a large proportion of the farmers interviewed:

> *"I stagger the planting over the week; twice in the course of the week . . . I only spread the planting over two days because of the limitations with labour availability. If I had a lot more people, I would plant the whole field in a day"* —Mr. PNK, male, 58, Asitey

Thus, while farmers know that the rains are unreliable, they appeared quite sure of when the time was right to commence sowing [67]. It is also pertinent to note that staggered sowing was inevitable where a household cultivates multiple maize fields at different locations. Multi-plot cultivation also necessitates the use of hired labour as household labour alone may not suffice for such time-sensitive activity as planting. This is illustrated by farmer who avers that:

*"We don't [intentionally] stagger planting, if one has the strength [labour power] . . . If you cultivate several acres, then you are going [have to] do it in multiple locations because you cannot get a large field in the same place. This means that you can plant a particular field today and then the other [field] the next day. But if you want all fields at the right time, then you have to rely on hired labour, which can be very expensive"* —Mr. AAA, male, 33, Akatawia.

What is interesting from the assertions by Mr. AAA above is the allusion that farmers could not rent several acres of farmland at the same location and that if one wanted to plant within the optimal planting period, not only would one have to rely on hired labour but also do so in multiple locations. This points to limitations relating to land availability and access and how these influence plot preparation in time for timely planting in resource-scarce contexts where voluntary labour is limiting. Apart from timing of planting, other important factors across villages and yield measures include household income levels, soil factors—average soil penetrability and total phosphorus content—level of fertiliser application, weed control and man hours contributed by voluntary labour.

### 4.2. Underlying Role of Socioeconomic Factors

Visual inspection of the aerial images of maize fields also show that dry patches are more pronounced around edges and borders of maize fields, especially those that are isolated from other maize fields and surrounded by fallowed and uncultivated farmlands. From ground-truthing these plots, dry patches have also been common under trees with large canopies. These last two findings are in line with those by Ndoli [68], who found that maize crop emergence and yields are severely affected due to competition with trees for nutrient and moisture. The interview data, among other things, shows that farmers rarely deliberately stagger or delay planting after the first rains and that labour limitation at the household level is a major impediment not only for timely plot preparation, but other important management activities such as planting, weed control and fertiliser application. This is in sync with the findings of Gianessi [29] that the timing of operations is strongly correlated with labour dynamics at the farm level.

On the back of the foregoing, two key underlying socioeconomic factors that deserve further attention are land tenure and labour dynamics. They are underlying because they influence management and soil factors, which in turn, more directly influence yields [32]. For the present study, while farmers are quite certain of the timing of the optimal planting window, planting often extends beyond this window because plot preparations have not been completed in time or they have multiple plots in separate locations due to limitations with land availability. Another important finding of the present study relates to the level of land availability and the implications that this has on the approach farmers tend to adopt in managing their plot. Here, the SLF becomes germane in understanding the differential asset portfolios of smallholders and how this in turn influences their differing livelihood strategies and outcomes [37]. Farmers who have the required human (active household members), social (volunteers from social groups) and/or financial (to hire labour) capitals would be able to carry out time-sensitive farm management activities timeously, while those who cannot would have to rely their own limited manpower [69]. Thus, decisions about maize planting dates in Central Ghana are often driven by other socioeconomic factors such as the availability of labour, capital or seeds [67].

With regards to limitations relating to labour availability, for example, an induced innovation is the increasing reliance on agrochemicals for plot preparation and weed control in the study area [47,48]. While the positives of herbicide usage in agriculture in terms of being relatively cheaper compared to hiring labour for weed control, ensuring timely weed control, reducing erosion and nutrient run-off has been touted [29], it can have serious environmental consequences when inappropriately used. The widespread application of herbicides not only for weed control during the season but even at the start of the season as part of the field preparation activities before planting is alarming. It requires sustained training of farmers through improved extension services on the types, timing, quantity and appropriate mixtures to be applied to minimise possible damage to crops and leaching into water bodies as well as ameliorate its negative impacts on maize yields.

An important corollary of the farmland scarcity is the reduction in average plot sizes and shorter fallow periods [70]. With regards to the former, the average maize field size of ~1 acre in both study villages is indicative of shrinking farm sizes and suggests that farmlands are increasingly scarce. A root factor of the phenomenon of farmland scarcity is the land tenure system in operation in the study communities. Here, like in most parts of Ghana where society is patriarchal in structure, farmlands are privately owned and usually passed on from fathers to male children [71]. Unmarried female offspring may, however, be allocated a smaller portion, but would have to relinquish the parcel or will it to younger children in the family at the point of marriage. Wives are expected to cultivate their husbands' lands Given the customary and economic importance that is attached to land, outright sale is rare; hence, the continuous division among offspring. Even when some siblings emigrate to urban centres and are not directly engaged in farming in their villages, they often hold onto their inheritance. The concluding part of the excerpt from the interview with Mr. AAA suggests that farmers who have the desire and ability to cultivate more than an acre are often forced to cultivate a number of plots at multiple locations. This often leads to delays in some important management activities such as planting in at least some of the locations. The timing of such time-sensitive activities has been shown to have critical implications in explaining yield levels [25,39]. Findings from the present study thus show that farmers do not deliberately stagger planting and that farmland scarcity significantly contributes to the delay in planting.

Furthermore, from the aerial images of maize fields, while the presence of poor patches on Asitey plots are largely haphazard, with the exception of the ploughed plots, poor patches on Akatawia plots are predominantly found along borders. Farmers explain this observation as deriving from the presence of large tree canopies at the centre of plots and thick vegetation surrounding their maize plots. They appear helpless largely due to the private ownership structure obtaining in these communities which restrict leaseholders from taking certain actions without the expressed permission of landowners. In Akatawia, however, the presence of such large tree canopies is limited to a maximum of a couple of Odum trees per plot. Such commercial trees are owned by landowners and only harvested when they encounter a sudden need for cash arising from emergencies such as funerals and serious illnesses of household members. Until such a need arises, leaseholders will have to cultivate their maize farms under the limitations imposed by the canopies of such trees. In Rwanda and Ethiopia, Ndoli [68] similarly found reduction in not only maize crop emergence but also yields due to competition for water and nutrients between crops and trees. The author, however, points out that trees can provide substantial income for the poorest households. This benefit is lost when the household cultivating the plot does not own the land. In such a situation, the yield reduction effect could be even more detrimental to household welfare.

Again, the SLF is useful in understanding the phenomenon of plot ownership structure and the attendant restrictions on the plot operators which is more instructive in Asitey, where farmers have the option of moving into the state-controlled lands managed by the Forestry Commission of Ghana. As Bebbington [36] posited, of all the resources available to individuals and households, access to common property resources is the most critical. Even more critical is the terms that govern this access. About 29% of the Asitey sample fall under this informal arrangement for accessing farm plots (Figure 2). These farmers do not pay any rent. They are allowed access to use these plots for a period on the foremost condition that they tend to teak trees planted by the Forestry Commission. Farmers are, acutely aware of the negative effects of tree shades, and the received literature shows that the presence of large tree canopies on maize plots is detrimental to crop vigour [72,73]. Farmers often try to ameliorate this by pruning teak branches. Farmers who are found to tamper with young teak trees and prune older ones are often summarily ejected from such lands given that their rights to these lands are informal. Farmers are, therefore, forced to modify their management activities by neglecting the crops around the teak trees. In seeking to praise the utility of the field health maps during the photo-elicitation interviews, for example, a farmer on one of such plots made the following instructive revelation:

*"These photos will be very useful to us because now we can clearly see [from the sky] which portions [of the maize field] are good and then we can neglect the portions that are not productive in the coming season. As you know, we don't pay for the land, it was released to us by the Forestry people so we will just concentrate on the productive parts and ignore the unproductive areas. The problem is that sometimes we are forced to continue clearing the whole plot because that is one of the conditions for having been given access to the land. If not, another farmer could start encroaching. So even if it is not yielding much maize, one could try other crops otherwise you lose your rights to the plot"* —Mr. JA, male, 77, Asitey.

What the excerpt above reveals is the willingness on the part of farmers in informal arrangements with officials of the Forestry Commission to abandon poor patches because they are not financially invested in terms of rent payment. Such farmers are also disincentivised from investing in yield improving interventions such as fertiliser application mainly due to the possibility of ejection at a moment's notice. Similarly, in spite of the ubiquity of its use, farmers who are outright owners often abstain from herbicide application out of concern for the long-term implications of its use on their lands. That is, rather than try to maximise yields for the present season, farmers who inherit and own their lands tend to prioritise yield optimization and stabilization. Such farmers, thus, rate long term sustainability of their lands higher than leaseholders who are more concerned about recouping their investments within the short term before their lease expires when the land reverts to original owners. For such farmers, yield maximization, even if short-term in outlook, is an attractive option. This is in line with the findings of Codjoe [74], that farmers in such situations tend to discount the future at very high rates, thereby reducing the incentive for long-term investments in improved soil fertility. Given this context, Benneh et al. [75] posited that sharecroppers in Ghana exert enormous pressure on soil fertility to secure high yields in order to pay land rents. The corollary, however, is nutrient mining given the intensive cropping that occurs without the necessary sufficient nutrient replacement through fertiliser application. This has contributed to more noticeable poor patches on such fields. In a land-abundant context, farmers can fallow their plots long enough for the natural regeneration process to be completed before coming back to such plots. However, in land-scarce settings, as obtained in the study area, farmers deal with this challenge by moving further into the hitherto uncultivated forest. Such younger plots generally stimulate more vigorous crops, as their soils are more fertile. Their downside, however, is that they tend to have trees with large canopies, which impede healthy crop growth within fields as well as the uncultivated parcels of land bordering them. It is on the latter kind of plots that poor patches tend to be dominant on the edges.

The second socioeconomic factor of interest were the labour dynamics at the household level, which fall under the human capital within the asset pentagon of the SL framework [35–37]. While limitations with farmland availability leads to multi-locations of maize fields, which in turn can contribute to a delay in planting, a household with an adequate supply of labour—either family or hired labour—may be able to still plant within the ideal planting window by preparing fields in time. As the excerpt from Mr. AAA also shows, the cost of labour is too expensive for most smallholder households to utilise regularly. Moreover, rural-urban migration of older children in search of white-collar jobs has contributed to the situation where the proportion of household population aged 16 years and younger is quite high. This means that the contribution of family labour to the overall labour requirements on the farm is largely limited. As Beza et al. [39] explains, family labour depends on household composition and off-farm opportunities, while hired labour depends on cash availability and labour market dynamics. These perhaps explain the proliferation of the use of herbicides for weed control rather than rely on manual methods, which require more labour, which can be expensive. The most viable alternative is thus voluntary labour, of which the importance is underscored by the MLR results in Table 3. From Table 3, the most important factors in the socioeconomic cluster of yield determinants are household income level and voluntary labour used for all farm activities. This finding is also instructive given the direct linkage between available income and a household's ability to hire

labour for crucial management activities, such as plot preparation, planting and weed control at the appropriate time [33].

*4.3. Importance of Multi-Methods and Data in Explaining Yield Levels*

An important distinguishing feature of the present study is our attempt at using multiple methods and data in an integrated manner to unravel present maize yield levels. The task has proven challenging in many respects. Perhaps the most important finding is that no one factor, or cluster of factors, adequately explain yield levels. Our integrated approach underscores the importance of approaching the analysis of yields from a holistic perspective. Smallholder farming, especially that which inheres in much of SSA, is characterised by complexity and heterogeneity and is a product of the environment in which farmers find themselves [76]. A nuanced understanding of the factors that impinge on farm productivity therefore requires that attention is paid not only to the environmental controls, but also socioeconomic, political and cultural milieus within which farmers operate [32]. Combining methods and data, as we have attempted to do here, is most revealing of not only the limitations of hitherto discipline-focused endeavours at understanding yield levels in SSA but also the inadequacy of factors or cluster of factors at explaining current yield levels. For example, from our MLR results, the most important factor—timing of planting—explains only 25% of the variance in CC yields in Akatawia, while timing of planting and household income level together explain 32% of the yields. This implies that only a third of the variance in the CC yields in Akatawia is explained. This also means that two-thirds of the yields is explained by other factors, even in our best performing model (Table 3). However, the use of the aerial photographs shows a prevalence of poor crop patches on the borders and edges of several of the maize fields (Figure 5). The interview data not only unravel the sources of these poor patches, but also explains why farmers tend to plant beyond the ideal planting window or with trees they do not derive any significant economic benefit from despite awareness of their implications on crop yields. Integrating data and methods thus brings to the fore the underlying role of socioeconomic factors relating to labour and land tenure dynamics that are often not given adequate consideration in studies that seek to explain yield levels [39].

## 5. Conclusions

The present study sought to employ an integrated approach to unravelling the sources of current yield levels on smallholder family farms in two farming villages in the Eastern region of Ghana. The integrated approach entailed the use of plot and household data, remote sensing data of maize fields and interview data on the aerial photos in a supplementary manner. We found that whichever yield measure was used, actual yields were far below what can be achieved even within the context of local limitations. This also shows the substantial scope that exists for yield improvements. However, these needed improvements in crop yields are not possible without first developing a holistic understanding of the controls of yields. Our attempt at an integrated approach—method- and data-wise—has shown that the factors that limit yields are varied and complex and cannot be adequately understood using a mono-disciplinary approach. From our analyses, none of the factors or category of factors adequately explain yield levels. While planting date relative to the first rains of the season has proven to be the most important factor influencing maize yields from the analyses of the survey data, spatial analysis of the aerial photographs of maize fields showed the concentration of poor crop patches on the borders and edges of fields. Incorporating the interview data shows that socioeconomic factors of labour and land tenure dynamics have an important underpinning influence on maize yields. Improving current poor yields will, therefore, require dealing with these socioeconomic dynamics.

With reference to land tenure dynamics, we found that rented fields are often continuously and intensively cultivated but without adequate replacement of crucial soil nutrients through appropriate fertilization levels. Additionally, given the poor tenure security associated with access to state lands, smallholders are not incentivised enough to invest already limited resources on the farm. Furthermore,

continuous division of farmlands through inheritance has led to small field sizes and the inevitability of cultivating in multiple locations where farmers are able and willing to cultivate a few acres. This comes with a number of complications, including delay in some time-sensitive farm activities. Those who seek to avoid this are often forced to move into lands that were hitherto uncultivated and further away from the village. Such fields, however, tend to lose crops on their borders and edges due to competition for nutrients, moisture and light from nearby bushes as well as the activities of birds and other rodents. The tenure issue is a complex one, and thus, will require major paradigm shifts in land reforms and administration to address land fragmentation and tenure security to engender on-farm investments.

**Author Contributions:** Conceptualization, I.W., M.J. and O.H.; data curation, I.W., M.J. and O.H.; formal analysis, I.W.; funding acquisition, M.J. and O.H.; investigation, I.W.; Methodology, I.W.; project administration, M.J. and O.H.; resources, M.J. and O.H.; software, I.W. and O.H.; supervision, M.J. and O.H.; validation, M.J.; visualization, I.W. and O.H.; writing—original draft, I.W.; writing—review and editing, M.J. and O.H. All authors have read and agreed to the published version of the manuscript.

**Funding:** This research was funded by the Swedish Research Council (Svenska Forskningsrådet Formas), grant number 2014-00646 and VR, grant number 04440.

**Acknowledgments:** We are grateful to the smallholder farmers in Asitey and Akatawia, who magnanimously allowed us to use portions of their maize farms for our field experimentations and surveys. We also acknowledge the help we received from local authorities and field assistants during data collection. We are grateful to the two anonymous reviewers for their critical and thorough reviews that helped shape the paper.

**Conflicts of Interest:** The authors declare no conflict of interest. The funders had no role in the design of the study; in the collection, analyses or interpretation of data; in the writing of the manuscript, or in the decision to publish the results.

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
