# Peer review of "An Integrated Approach to Unravelling Smallholder Yield Levels: The Case of Small Family Farms, Eastern Region, Ghana"

_agriculture, doi:10.3390/agriculture10060206_

Round 1

Reviewer 1 Report

In the current version, most of the shortcomings indicated in the review have been removed, except for note no. 3:
"It is necessary to improve the interpretation of the results obtained from multiple regression models. The authors used the obtained regression coefficients (after the standardization of variables) to determine the strength of the influence of individual variables. There are no reservations to this, however , the square of the regression coefficient equals the determination coefficient only in the case of a regression with one explanatory variable. In the case of multiple regression, this is not true. "

The authors still state:
The timing of planting is most impactful in model 3 - R2 = 0.50

While 0.50 is one of the reported multiple regression coefficients.

Author Response

We would like to thank the anonymous reviewer for the thorough review. Further interpretation of the multiple linear regression has been added in L453-461 as follows: "For model 3, the MLR predicts timing of planting relative to onset of rains, soil PC 3 (% of silt and soil electrical conductivity), maize planting density, and household income level as significant factors. A significant regression equation was found (F(2,42) = 9.782, p<.000), with an R2 of .32. Predicted CC yields is equal to 4,563.65 – 337.5 (MAIZE PLANTING TIME) – 426.94 (INCOME GROUP), where maize planting time is measured in weeks after first major rains and coded as 1=1 week after rains, 2 = 2 weeks after rains, 3 = 3 weeks after rains, etc. and income group coded as 1 – 0-100$, 2 = 101-200$, 3 = 201-300$. Thus, CC yields in Akatawia reduced 337.59 kg for every week’s delay in planting and $100 reduction in household income".

Reviewer 2 Report

The work used an integrated approach to investigate the sources of current crop yield levels and their variability at the plot levels in two maize farming villages in the Eastern region of Ghana. A multi-scale sampling strategy was deployed. The paper used data collected from the two study villages in a cross-sectional comparative mixed-methods framework.

There was an accurate investigation of problems presented, a full description of methods used, and a good interpretation od data.

Best regards.

Author Response

We note no additional comments/concerns from Reviewer 2 respond to. We would like to express deep appreciation to the two anonymous reviewers for the reviews and helpful comments.

This manuscript is a resubmission of an earlier submission. The following is a list of the peer review reports and author responses from that submission.

Round 1

Reviewer 1 Report

Summary:

Adding to the wide body of litterature on yield gap analysis, this paper is a piece of work analyzing yield variability in smallholder context (Ghana), using several approaches ranging from statistical to remote sensing image analysis. While the approach could be potentially very interesting and novative comparing to what it is traditionnaly done on that topic, the key message, the narrative thread and the methodological framework are finally get lost in details that serve badly the paper. This work need to be considerably rework before to be published.

Specific comments :

Abstract :

L19. Here and latter in the manuscript, the reference to the SLF is not clear. To what extent your approach is included an or contributes to the SLF ? I’m not convinced by this reference to the SLF and it brings nothing to your study. I have a partial understanding of SLF but I have the feeling that your study contributes only to a marginal part to SLF. Maybe I’m wrong but authors have to better justify this part otherwise I would suggest to remove it from their manuscript.

Introduction :

L139-140. I’m not fully agree with that statement. Causes and drivers of yield variability are relativement well known when considered individually. By the way, you mention some of the causes in your introduction. What need to be strenghten however, is our understanding of how these different drivers interact together (linear, non linear, additive …?), what are the hierarchy between them, at what relevant scale. This study should be more position on that way.

In addition, what do you mean by micro-level ? The plot level as said after ? Sometimes you refer to micro level, sometimes to micro scale. However, scale and level have a different semantic : scale is the extent of the land area considered while the level is the unit of observation.

L145. After reading the paper, I don't understand what do you call an integrated approach? In the present case, It looks like more as a juxtaposition of approaches without link between them.

Material and methods :

Figure 1. Background of the map probably comes from OpenStreetMap! The source of the background should be specified in the legend.

L188-189. I don’t understand what is the role of the qualitative work here? How it is used after ? In addition, it was conduct 3 years after the quantitative work. If the aim is to strenghten the quantitative analysis, you introduced a lot of incertainity in our work by using information collected 3-years after.

L189. Here again, I don’t understand what multi-levels stands for ?

L261-2622. Very basic things, not needed.

L265-266. Why restrincting the PCA only to soil variables ? An option would be to conduct a PCA on the full core of variables to keep the most explaining variables in the final MLR model while limiting overfitting. In your MLR model, a part for the soil variables, you didn’t test for any colinearity among covariates (eg. VIF value), which is an issue.

L268. Given that relationships among the targeted variables (yields) and the covariates in one side, and between the covariates in other side are intuitively far from being linear, I’m not sure that a MLR is really a good option to answer your question. Up-to-date machine learning and non parametric algorithm as Random Forest or GBM would be more interesting to test here. They allow to handle different types of independent variables and are able to fit complex non-linear relationships and interactions effects between independent variables. In addition, to emphasize the main factors of yields variability you can rely on the relative influence (or contribution) measure of each independent variable. I guess It would give a real adding value to your work.

L285. Why gNDVI instead of traditionnal NDVI? what is the adding value of using gNDVI? As it relies on the green wavelength, gNDVI is more sensitive to nutrient deficiency but if the objective is to have an overview of the general crop health , NDVI is more designed,

Results

This section need to be considerably reworked. It is too long, with too many unecessary description and details. The main results are get lost in details. Please focus only on the main results with tangible facts and figures.There is a lot of descriptive and informative things that would better fit to the discussion section.

L306. The authors need more transparency on what they did. This is really problematic here because a lot of figures are give but without any tables to support them. Please add a table with at least the main statistics and associated pvalue for comparison between sites.

Figure 2. Please remove the grey background, it makes heavy the figure.

L368. PCA tests have to be presented in the method section.

L439. This section is too long. The result section has to be concised and precised.

L453-455. Two maps are not comparable since they are not on the same scale. In plot B you have negative value (more stressed pixels) that is not the case for plot A ... You should have both plot on the same scale.

L455-457. There is a little adding value to perform a spatial comparison of plots. Even if you try to limite heterogeneity between plots (ie same management etc), they are not in the same biophysical environment.

Plate 1. Why plate ? It is a figure isn’t ? Please make both map on the same color scale.

Plate 2. No legend ? No geographical scale ? No orientation ? The color scale should be the same as in Plate 1. Again this is a figure.

Discussion

L576. Several publications on the importance of planting windows on cereal crops yields in smallholder agriculture over west africa are missing.

Marteau, R., Sultan, B., Moron, V., Alhassane, A., Baron, C., Traoré, S.B., 2011. The onset of the rainy season and farmers’ sowing strategy for pearl millet cultivation in Southwest Niger. Agric. For. Meteorol. 151, 1356–1369. doi:10.1016/j.agrformet.2011.05.018

Waha, K., Müller, C., Bondeau, A., Dietrich, J.P., Kurukulasuriya, P., Heinke, J., Lotze-Campen, H., 2013. Adaptation to climate change through the choice of cropping system and sowing date in sub-Saharan Africa. Glob. Environ. Chang. 23, 130–143. doi:10.1016/j.gloenvcha.2012.11.001

Bacci, L., Cantini, C., Pierini, F., Maracchi, G., Reyniers, F.N., 1999. Effects of sowing date and nitrogen fertilization on growth, development and yield of a short day cultivar of millet (Pennisetum glaucum L.) in Mali. Eur. J. Agron. 10, 9–21. doi:10.1016/S1161-0301(98)00046-X

Srivastava, A.K., Mboh, C.M., Gaiser, T., Webber, H., Ewert, F., 2016. Effect of sowing date distributions on simulation of maize yields at regional scale – A case study in Central Ghana, West Africa. Agric. Syst. 147, 10–23. doi:10.1016/j.agsy.2016.05.012

L620-622. Not needed, already said in the precedent section.

L626-627. Sure, but it is strongly linked to the tree species within and outside the fields. Are the tree species you have comparable to the one of Ndoli et al. ?

Conclusion.

L789-811. It is more discussion elements than conclusion elements.

Reviewer 2 Report

The paper presents a fairly novel approach to looking for determinants of maize yields.
However, some elements could be improved.
1. Description of the sample selection, the authors could briefly describe the methodology they refer to, without forcing the reader to search for the text "The Millennium Development Goals and the African Food Crisis - Report from the AFRINT 922 II project".
2. It seems necessary to add a full description of the multiple regression models used, along with a brief description of the variables used (maximum, minimum, average, standard deviation).
3. It is necessary to improve the interpretation of the results obtained from multiple regression models. The authors used the obtained regression coefficients (after the standardization of variables) to determine the strength of the influence of individual variables. There are no reservations to this, however, the square of the regression coefficient equals the determination coefficient only in the case of a regression with one explanatory variable. In the case of multiple regression, this is not true.
4. If there are such pronounced shading effects by trees, why will the authors not add the percentage of the shaded area as an explanatory variable?
General remark about the regression model. In the case of crops, the principle of limiting factor applies. For example, if there is a shortage of water, using high doses of nitrogen will not increase the yield. Similarly, extensive varieties will not respond to increased fertilization. The authors should check more carefully which factors limited the yield because it may be the reason for the irrelevance of fertilization and generally low yields.